# Raptors avoid the confusion effect by targeting fixed points in dense aerial prey aggregations

Caroline H. Brighton [1] ✉, Laura N. Kloepper[2,3], Christian D. Harding [1,4], Lucy Larkman[1], Kathryn McGowan[2], Lillias Zusi[2] & Graham K. Taylor [1] ✉

Collective behaviours are widely assumed to confuse predators, but empirical support for a confusion effect is often lacking, and its importance must depend on the predator's targeting mechanism. Here we show that Swainson's Hawks *Buteo swainsoni* and other raptors attacking swarming Mexican Free-tailed Bats *Tadarida brasiliensis* steer by turning towards a fixed point in space within the swarm, rather than by using closed-loop pursuit of any one individual. Any prey with which the predator is on a collision course will appear to remain on a constant bearing, so target selection emerges naturally from the geometry of a collision. Our results show how predators can simplify the demands on their sensory system by decoupling steering from target acquisition when capturing prey from a dense swarm. We anticipate that the same tactic will be used against flocks and schools across a wide range of taxa, in which case a confusion effect is paradoxically more likely to occur in attacks on sparse groups, for which steering and target acquisition cannot be decoupled.

Dense aggregations of flocking, schooling, and swarming animals are widely assumed to confuse predators, but evidence for a confusion effect is far from universal[1]. Other mechanisms reducing the predation risk of grouped prey depend closely on predator behaviour. Prey individuals will only benefit from attack abatement[2–4] if a predator's attack rate scales less than proportionally with group size, which depends on its search strategy and feeding efficiency. Theories of marginal predation[5,6] and the selfish herd[7–9] are likewise underpinned by the predator's tendency to attack nearby individuals[10], and even the shared benefits of group vigilance[11] may depend on its mode of attack[12]. The occurrence and significance of any confusion effect must similarly depend on the predator's attack behaviour[1]. For example, it is reasonable to expect confusion to arise in raptorial predators that single out individuals, rather than in engulfment predators that target the group, but the detail of how the attacker homes in on its target is presumably decisive. Here we analyse the visually guided flight behaviour of hawks attacking swarming bats emerging in massive column formation from their roost (Movie S1), using three-dimensional (3D) video reconstruction techniques and behavioural algorithms to model their behaviour.

Confusion is supposed to occur when the presence of many targets compromises a predator's ability to target an individual[13], particularly if those targets are similar in appearance or moving in a coordinated fashion. Yet, empirical support is often lacking—partly because of the difficulty of designing experiments that eliminate other confounding factors[13–16]. Bats emerging from their roosts often form massive and highly dynamic swarms that present many sensory challenges to their predators (Fig. 1). For instance, although the procession of the swarm may be quite coherent, individual movements are often highly erratic (Fig. 1A) producing a tangle of interwoven flight paths (Fig. 1B). Such protean behaviour makes individuals particularly difficult to track, and may be expected to amplify the confusion effect in a dense group[17]. Nevertheless, a previous observational study of the model system that we consider here found that Swainson's Hawks

[1]Department of Biology, University of Oxford, 11a Mansfield Road, Oxford OX1 3SZ, UK. [2]Department of Biological Sciences and Center for Acoustics Research and Education, Spaulding Hall, University of New Hampshire, Durham, NH 03824, USA. [3]Department of Biology, Saint Mary's College, 262 Science Hall, Notre Dame, IN 46556, USA. [4]Department of Physiology, Anatomy, and Genetics, University of Oxford, Sherrington Building, Parks Road, Oxford OX1 3PT, UK. ✉e-mail: caroline.brighton@biology.ox.ac.uk; graham.taylor@biology.ox.ac.uk

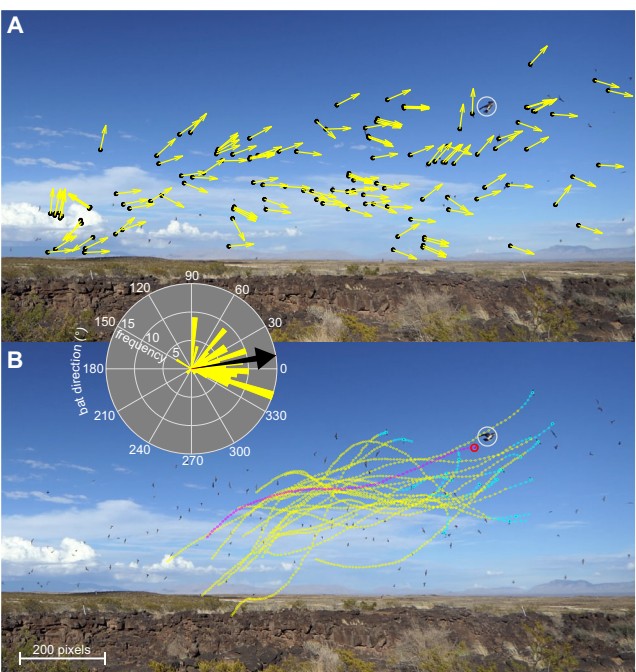

**Fig. 1 | Swarming behaviour of Mexican Free-tailed Bats during an attack by a Swainson's Hawk. A** Instantaneous flight direction (yellow arrows) of bats (black points) tracked during an attack by an incoming hawk (white circle), as seen from a fixed video camera. Not all bats are tracked, but the inset polar plot summarises the distribution of the projected flight directions from (A), with the black arrow indicating the circular mean. The bats' motion is coherent in the sense that the swarm moves consistently from left to right, but displays significant variability reflecting protean behaviour within the swarm. **B** Flight trajectories of 20 bats (yellow points) leading up to the catch attempt in the frame shown; flight trajectories are sampled at 0.02 s spacing, with the last 5 points highlighted in cyan. The trajectory of the bat (red circle) attacked by the hawk (white circle) is shown in magenta. Note that the multiple interwoven flight paths make it exceedingly difficult to track individuals—even with the benefit of being able to move back and forth through a sequence of video frames. Tracking is even more challenging in real time, so the bats' protean behaviour could be expected to amplify the confusion effect arising from having many similar individuals visible simultaneously. All frames shown are from the original video collected by the authors in this study.

*Buteo swainsoni* hunting Mexican Free-tailed Bats *Tadarida brasiliensis* in broad daylight (Movie S1) were no less successful when attacking bats flying within the column rather than when attacking bats flying apart from it[18]. How were the hawks able to avoid confusion when attacking the swarm, and what are the broader implications for our understanding of the confusion effect in attacks on dense prey aggregations?

Attacks on solitary prey require accurate guidance. Open-loop interception may be possible against a predictably moving target, and means that the attacker aims at its target's expected future position, without making further corrections to account for unexpected changes in the target's relative motion (e.g. an archer shooting a moving target with a bow only has control over the arrow's initial release velocity). On the other hand, closed-loop steering is essential when pursuing an evasive target, and means that the attacker uses information on the relative motion of its target to make continuous adjustments to its own flight trajectory (e.g. a heat-seeking missile intercepts its target by tracking its thermal signature). The quantitative mapping from the relative motion of a target to the steering response of its attacker is called a guidance law[19]. Fitting guidance laws to biological data has provided quantitative insight into the simple behavioural algorithms[20] that describe how raptors catch lone prey[21–23]. For example, the terminal attack trajectories of Peregrine Falcons *Falco*

*peregrinus* and Gyrfalcons *F. rusticolus* attacking lone targets are well modelled by a classical missile guidance law[19] called proportional navigation[22,23], whereas the attack behaviours of Harris' Hawks *Parabuteo unicinctus* are better modelled by a mixed guidance law[21].

Proportional navigation (PN) commands the attacker's turn rate $\dot{\gamma}$ (i.e. the rate of change of the bearing $\gamma$ of its velocity) in direct proportion to the target's line-of-sight rate $\dot{\lambda}$ (i.e. the rate of change of the bearing $\lambda$ to its target). Pure PN guidance, $\dot{\gamma}(t) = N\dot{\lambda}(t - \tau)$ where $N > 0$ is the navigation constant and $\tau \geq 0$ is a time delay, is most effective against non-manoeuvring or smoothly manoeuvring targets, and works well in the open environments typical of falcons[22,23]. In contrast, the pursuit trajectories of Harris' Hawks are best modelled by augmenting PN with a proportional pursuit (PP) term that commands turning at a rate proportional to the deviation angle $\delta$ between the attacker's velocity vector and its target[21]. This mixed PN + PP guidance law, $\dot{\gamma}(t) = N\dot{\lambda}(t - \tau) - K\delta(t - \tau)$ where $K > 0$ is another guidance constant, outperforms PN against erratically manoeuvring targets[21]. Furthermore, by promoting tail-chasing, mixed PN + PP guidance may reduce collision risk in the cluttered environments that hawks frequent. For ease of interpretation, we have shown these guidance laws in a scalar form applicable to planar pursuit, but each is readily generalized to a vector form applicable to three-dimensional (3D) motion, and they are analysed as such below (see Methods).

To our knowledge, previous algorithmic studies of pursuit-evasion have only analysed attacks on solitary targets, so it remains unknown what guidance laws—if any—predators use when attacking aggregated prey. In principle, open-loop interception could be more reliable against dense prey aggregations, because of the elevated probability of achieving a hit by chance with multiple targets present. This could in turn have important implications for the occurrence and significance of any confusion effect, and hence for the functional mechanisms promoting collective behaviour. Here we test whether the behavioural algorithms that aerial predators use to attack dense swarms are the same as those used to attack lone prey. We go on to show how the different targeting behaviours used when attacking dense swarms avoid the confusion effect that is widely assumed to be one of the key mechanisms by which individual prey benefit from collective behaviour.

## Results

We recorded attacks over 21 days, using three pairs of high-definition video cameras fixed in stereo configuration around the cave from which the bats emerged (Movie S1). We then tracked the hawks and the individual bats that they caught or attempted to catch (Fig. 1), which allowed us to reconstruct their 3D attack trajectories (see Methods). We reconstructed $n = 62$ terminal attack trajectories in this way, and another $n = 26$ long-range approaches in which we were able to track the hawk but not the individual bat that it attacked. We also reconstructed $n = 2$ long-range approaches recorded for a Peregrine Falcon or Prairie Falcon *F. mexicanus* that attacked the swarm on two consecutive evenings. We analysed these flights by finding the values of the guidance constants $N$ and/or $K$ that minimized the mean absolute distance ($\eta$) between the observed and simulated data for each flight, in numerical simulations commanding steering according to the vector form of the mixed PN + PP guidance law $\dot{\gamma}(t) = N\dot{\lambda}(t - \tau) - K\delta(t - \tau)$, where we set $K = 0$ or $N = 0$ to fit pure PN or PP respectively (see Methods). Other things being equal, longer simulations have greater potential to diverge from the observed flight track, and we therefore report the relative error $\varepsilon = \eta/l$, where $l$ denotes the total path length.

### Terminal attack trajectories provide no evidence of closed-loop pursuit within the swarm

We began by fitting the three candidate guidance laws to the $n = 62$ terminal attack trajectories that we recorded from the hawks, under

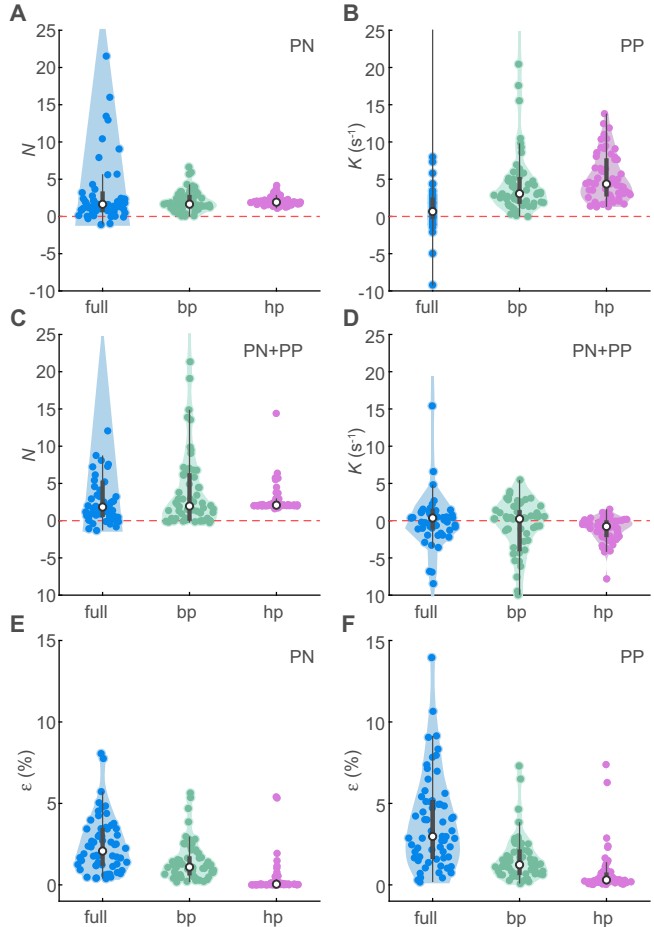

**Fig. 2 | Summary of guidance models fitted independently to _n_=62 terminal attack trajectories of Swainson's Hawks attacking Mexican Free-tailed Bats.** Violin plots showing kernel density estimates with data points overlaid, comparing (**A**–**D**) the fitted guidance parameters _N_ and/or _K_ and (**E**, **F**) relative error _ε_ for the three alternative target definitions (full, blue: instantaneous bat position; bp, green: final bat position; hp, magenta: final hawk position) and three candidate guidance laws: (**A**, **E**) PN, (**B**, **F**) PP, and (**C**, **D**) PN + PP. White circle denotes median; thick grey bar represents interquartile range; thin black line indicates outlying points falling >1.5 times the interquartile range below the first quartile or above the third quartile. Parameter estimates for _N_ and _K_ did not always converge on reasonable values for attacks involving little turning; panels (**A**–**D**) do not display some datapoints with higher values of _N_ and _K_, but all datapoints are used to compute the kernel density estimates and the median and interquartile range. Source data are provided as a Source Data file.

the assumption that they tracked the individual bat that they attempted to catch, without delay at $\tau = 0$ (Table S1). The median guidance parameter estimates for $N$ and $K$ were positive as expected under pure PN or PP guidance ($\widetilde{N} = 1.6$, CI: 1.1, 2.0; $\widetilde{K} = 0.7$, CI: 0.1, 1.7 s$^{-1}$; Fig. 2A, B), but the parameter estimates for $K$ were inconsistently signed under mixed PN + PP guidance ($\widetilde{N} = 1.8$, CI: 1.2, 2.6; $\widetilde{K} = 0.4$, CI: −0.3, 0.7 s$^{-1}$; Fig. 2C, D), which suggests that they were overfitted. Of the two pure guidance laws, PN fitted the data more closely than PP (two-tailed sign test: $p = 0.007$; $n = 62$; Fig. 2E, F). We obtained qualitatively similar results with delays $0 < \tau \leq 0.1$ s but found that the error was minimized at $\tau = 0$. We conclude that PN is the best supported of the three candidate guidance laws, conditional upon the assumption that the hawks steered after whichever bat they attacked. We tested this assumption by asking whether the hawk's steering was influenced by the bat's trajectory up to the point at which the hawk extended its legs in a grab manoeuvre[18], but we found no evidence that this was so. On the contrary, delay-free PN simulations treating the bat's final

position as the target of the hawk's guidance ($\widetilde{N} = 1.7$, CI: 1.5, 2.1) fitted the data more closely (two-tailed sign test: $p < 0.001$; $n = 62$; Fig. 2E) and with lower median relative error ($\widetilde{\varepsilon} = 0.011$; CI: 0.009, 0.013) than those targeting its instantaneous position ($\widetilde{\varepsilon} = 0.021$; CI: 0.017, 0.027). We therefore find no evidence that the hawks engaged in closed-loop pursuit of the bat that they attacked. As a check on the robustness of this conclusion, we applied an analogous model selection approach to published data on Peregrines and Gyrfalcons pursuing singleton targets[22,23], which confirmed as expected that their attack trajectories were better modelled by delay-free PN targeting the instantaneous rather than final position of the target (Table S2). As this is the opposite of what we observed for hawks attacking swarming targets here, it is reasonable to test whether the hawks' observed turns are instead consistent with the use of some form of open-loop steering behaviour.

### Terminal attack trajectories approximate constant radius turning into the swarm

The simplest possible model of turning involves the hawk making a constant radius turn into the swarm. This behaviour can be simulated using the same algorithmic approach as before, because delay-free PN has the property[19] of generating a constant radius turn towards a stationary target if the navigation constant is fixed at $N = 2$. Our finding that the hawks' terminal attack trajectories were well modelled by PN targeting the bat's final position at median $\widetilde{N} = 1.7$ (CI: 1.5, 2.1) could therefore be an artefact of constant-radius turning. We tested this interpretation by fitting delay-free PN targeting the final position of the hawk rather than the final position of the bat, which yielded navigation constants even closer to the theoretical value of $N = 2$ producing a constant radius turn ($\widetilde{N} = 1.9$, CI: 1.8, 2.0; Fig. 2A). As a final check, we fixed the navigation constant at $N = 2$ in delay-free PN treating the final position of the hawk as a virtual target (Fig. 3). The resulting simulation model has no free parameters, but still modelled the hawks' terminal attack trajectories more closely (two-tailed sign test: $p < 0.001$; $n = 62$) and with lower median relative error ($\widetilde{\varepsilon} = 0.003$; CI: 0.001, 0.008) than the best of the fitted guidance models targeting either the bat's trajectory or its final position (Fig. 2E). It follows that there is no evidence for the use of closed-loop pursuit over even the simplest possible model of open-loop steering behaviour matching the hawk's initial flight velocity and subsequent flight speed. In summary, our data provide no evidence for the use of closed-loop pursuit, and are more compatible with the hypothesis that the hawks turned on an approximately constant radius into the swarm, before grabbing at whichever bat they found themselves on a collision course with.

### Attacked bats remain on a near-constant bearing during the terminal attack trajectories

A striking feature of the hawks' terminal attack trajectories is that their line-of-sight to the bat remains nearly parallel throughout (Fig. 3). This constant-bearing geometry holds for any pair of objects on a collision course, so is inevitable during the final moments of any successful attack, but could nevertheless simplify the problem of singling out a bat to catch (see Discussion). To verify whether this constant-bearing geometry was particular to the individual bats that the hawks attempted to catch, we therefore plotted the azimuth and elevation of their line-of-sight for all $n = 62$ terminal attack trajectories (Fig. 4A, B), together with those of 20 neighbouring bats tracked concurrently over each of four attacks on which the bats were scattered enough to be individually identified across camera views (Fig. 4B, C). These polar plots display a radial pattern demonstrating the near-constant azimuth and elevation of the line-of-sight for the attacked bats (Fig. 4A, B), but show a much less radial pattern for other neighbouring bats (Fig. 4C, D). The circular standard deviation of the line-of-sight angle was significantly smaller in azimuth (two-tailed Wilcoxon rank sum test: $p = 0.008$) for the $n = 62$ attacked bats (median: 0.19˚; Q1, Q3: 0.11, 0.24˚) than for the $n = 80$

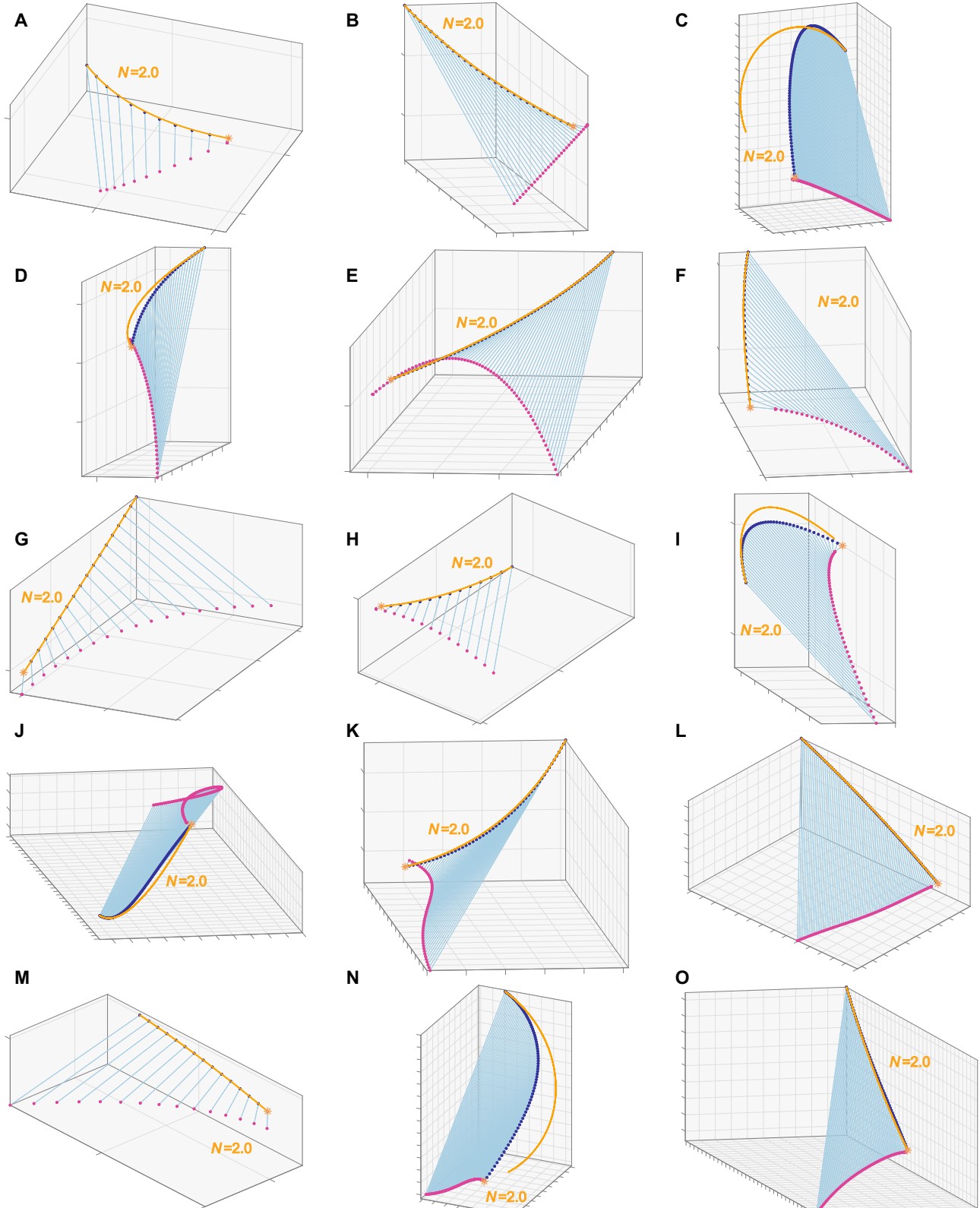

**Fig. 3 | Terminal attack trajectories of Swainson's Hawks successfully attacking Mexican Free-tailed Bats. A–O** Each panel plots the reconstructed three-dimensional trajectory of a hawk (dark blue points) capturing a bat (magenta points), connected by the instantaneous line-of-sight from the hawk to the bat (blue line), for all $n = 15$ attacks ending in a successful capture; orange starburst shows the point of capture. The line-of-sight remains nearly parallel through time, because the hawk and bat are on a near-collision course (see Discussion; Figs. 4, 6). Orange lines plot simulations of the hawk's flight trajectory generated under delay-free PN guidance at a fixed navigation constant of $N = 2$, treating the final position of the hawk as the target. This serves to generate a constant radius turn that satisfies the kinematic constraint of passing through the hawk's initial and final positions whilst also matching its initial flight velocity. Flights involving steep dives or climbs are not usually well modelled as a constant radius turn (e.g. **C, I, N**), but other flights are modelled quite closely. Grid spacing: 1 m.

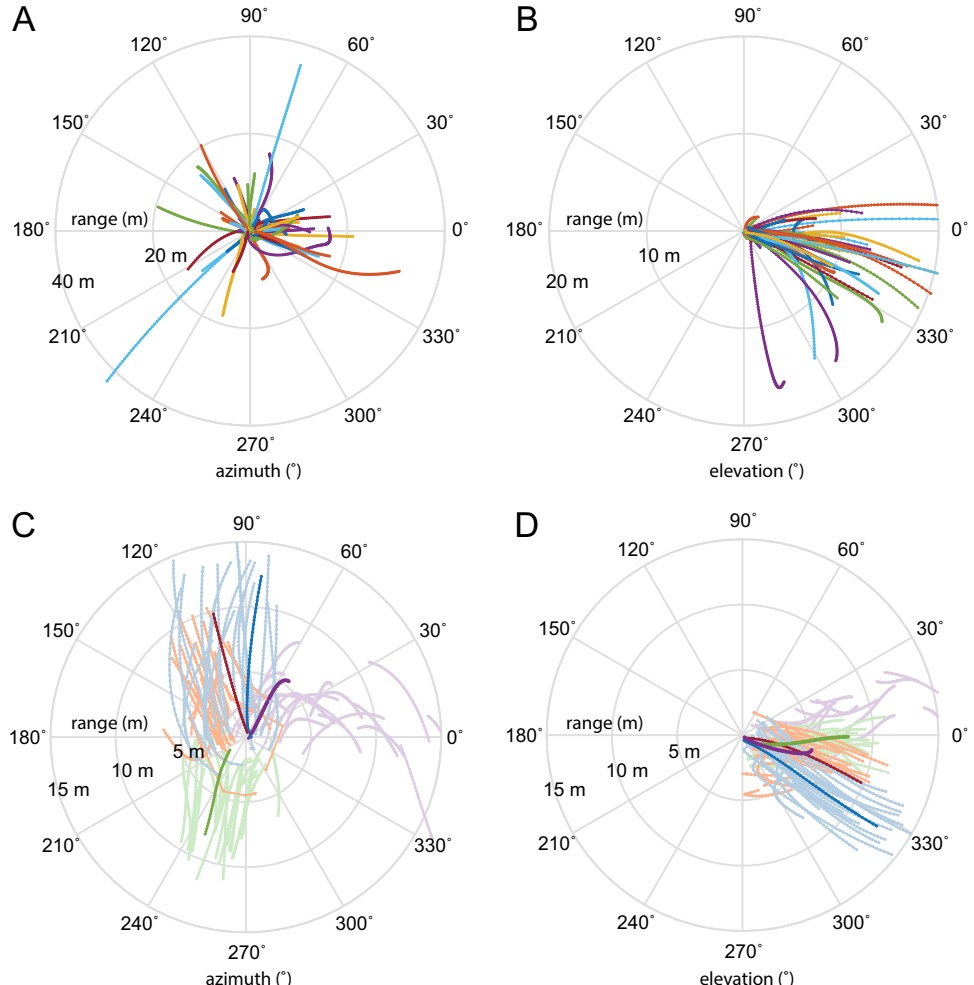

**Fig. 4 | Constancy of line-of-sight angle during the terminal attack trajectories of Swainson's Hawks attacking Mexican Free-tailed Bats. A, B** Polar plots displaying the azimuth and elevation of the line-of-sight (LOS) from the hawk to the bat that it attempted to catch for all $n = 62$ attacks; the radial coordinate represents the bat's range. The plotted lines are nearly radial, indicating that the attacked bat remains on a nearly constant bearing to the hawk. **C, D** Polar plots displaying LOS azimuth and elevation for 20 neighbouring bats tracked concurrently on each of

four colour-coded attacks; darker lines display the LOS azimuth and elevation of the bat that was attacked. These darker lines are more nearly radial than the lighter lines displaying the LOS azimuth and elevation of the neighbouring bats, confirming that the constant-bearing geometry is specific to the attacked bat. Only one of these four flights (purple) ended in a successful capture: this attack displays the most nearly radial of the dark lines; this being the flight on which the collision conditions are most closely met.

neighbouring bats (median: 0.28˚; Q1, Q3: 0.13, 0.37˚). The circular standard deviation of the line-of-sight angle did not differ significantly between groups in elevation (two-tailed Wilcoxon rank sum test: $p = 0.57$), being small for both the attacked individuals (median: 0.13˚; Q1, Q3: 0.08, 0.18˚) and the neighbouring bats (median: 0.15˚; Q1, Q3: 0.06, 0.21˚). This constant-bearing decreasing-range geometry was therefore a reliable feature of the attacked bats, which could potentially be used by the hawk to distinguish them from neighbouring bats during the terminal phase of the attacks (see Discussion).

**Long-range approaches approximate constant radius turning into the swarm**

The terminal attack trajectories that we modelled were necessarily quite short (median path length: $\tilde{l} = 6.7$ m; Q1, Q3: 2.8, 11.8 m), as they were limited to the brief interval over which it was possible for us to track the bat that the hawk attacked (median duration: 0.6 s; Q1, Q3: 0.3, 1.0 s). To test the range of distances over which the hawks' steering behaviour could be well approximated as a constant radius turn, we therefore simulated delay-free PN targeting the hawk's final position at a fixed value of $N = 2$ for the 26 long-range approaches (Fig. S1). We ran

these simulations beginning from different points in the flight, from 1.0 s up to a maximum of 20.0 s before the grab manoeuvre[18], in 0.2 s intervals. We identified the longest simulation for which the relative error was $\varepsilon \le 0.012$, this being the same error tolerance used in previous studies[22,23]. We found that we could model 20 of the $n = 26$ long-range approaches under delay-free PN at $N = 2$ to within this 1.2% error tolerance, with a median path length of 24.5 m (Q1, Q3: 11.9, 45.6 m) and a median duration of 2.1 s (Q1, Q3: 1.4, 3.9 s). For comparison, we also applied the same modelling approach to the two long-range approaches that we recorded from a Peregrine or Prairie Falcon. Simulating delay-free PN guidance at a fixed value of $N = 2$, we found that we could model both falcon trajectories at 1.2% error tolerance over distances of 48.4 and 70.6 m and durations of 4.0 and 4.4 s (Fig. S2). It follows that most of the long-range attacks that we observed could be well approximated as constant radius turns initiated well before the hawk had entered the swarm.

**Proportional navigation is a plausible mechanism for targeting a fixed location within the swarm**

Our simulations targeting a fixed point in space under PN guidance at $N = 2$ provide an algorithm solving for the constant radius turn that

uniquely satisfies the kinematic constraints of matching the initial conditions and intercepting the target. It can be shown theoretically[19] that the resulting flight path describes a curve of radius $(2\sin\delta_0)/r_0$ where $\delta_0$ is the initial deviation angle and $r_0$ is the initial distance to the target. Hence, whilst it is possible in principle that an attacker could calculate the radius of turn needed to reach a fixed point in space, doing so requires an estimate of target range, which may be lacking given the absence of any evidence for stereopsis in diurnal raptors[24]. The same will hold true of any prediction model in which the future position of an individual target is predicted explicitly. We are therefore left with two alternative hypotheses that are equally consistent with the data: first, that the hawks enter a turn of approximately constant radius, with or without knowledge of where they will intercept the swarm; second, that the hawks use PN guidance at $N \approx 2$ to target a fixed point in space, without knowledge of what radius of turn they will follow. These hypotheses each represent very different mechanisms: whereas the first describes a form of open-loop steering behaviour, with or without prediction, the second describes a form of closed-loop guidance from which a similar-looking behaviour emerges.

As a direct check on the validity of the second hypothesis, we tried modelling the long-range approach flights using delay-free PN at the best-fitting value of $N$ for each flight, treating the bird's final position as a virtual target (Fig. 5). This allowed us to model 24 of the $n = 26$ hawk approaches to within the same 1.2% relative error tolerance as before, with a median path length of 28.8 m (Q1, Q3: 14.3, 43.8 m), and a median duration of 2.5 s (Q1, Q3: 2.0, 4.6 s). The median guidance parameter estimate in these models ($\tilde{N} = 1.9$, CI: 1.6, 2.4) was statistically indistinguishable from the theoretical value of $N = 2$ corresponding to a constant radius turn. Hence, whilst there is clearly a risk of over-fitting associated with estimating the best-fitting value of $N$ independently for each flight (Fig. 5), allowing this to vary between flights extends the total distance of flight modelled at 1.2% error tolerance by just over one fifth for the hawks, and by more than a half for the falcons (Fig. S2). In summary, the long-range approaches that we observed are consistent with the hypothesis that the birds steered open-loop into the swarm on a turn of approximately constant radius, but can be even more closely modelled under the hypothesis that they used PN guidance to steer closed-loop towards a fixed point in space within the swarm.

### Hawks display no evidence of confusion and no evidence of preferentially targeting lone bats

Of the $n = 62$ terminal attack trajectories that we recorded from Swainson's Hawks, eight involved attacks on lone bats flying outside of the column (Fig. 6A). This proportion (13%; CI: 6, 24%) is consistent with the results of an earlier study of the same model system using other data[18], which found that lone bats were attacked at disproportionately high frequency (10%; CI: 7, 15%) compared to their overall representation in the population (-0.2%), such that bats flying within the column benefitted from attack abatement (Fig. 6B). Moreover, as in the previous study[18], we found no significant difference in the success rates of attacks on lone bats versus attacks on the swarm column (two-tailed Fisher's exact test: odds ratio: 2.1; CI: 0.4, 10.1; $p = 0.39$; $n = 62$), at observed success rates of 38% and 22% respectively (Table S3; Fig. 6). This earlier study left unresolved whether the higher attack rates observed against lone bats resulted from the hawks having an individual preference for targeting lone bats, or from a statistical tendency for the hawks to encounter marginal individuals first[18]. Our finding that the hawks showed no evidence of steering after the bats that they attacked strongly suggests that the hawks were not targeting individual bats at all. Indeed, even for the subset of attacks made on lone bats, the hawks' trajectories were more closely fitted by PN targeting the hawk's final position at a fixed value of $N = 2$ than the bat's instantaneous position (two-tailed sign test: $p = 0.008$; $n = 8$). Hence, although we cannot exclude the possibility that the hawks computed

an open-loop collision course to enable them to intercept these lone bats, it is more parsimonious to suppose that they encountered them at random, and that the higher predation risk experienced by lone bats simply reflects the fact that marginal individuals are likely to be encountered first when turning into the swarm.

## Discussion

Whereas the question of how aerial predators guide their attacks on singleton targets has been the subject of multiple studies[21–23,25], the question of how they guide their attacks on agile swarming targets does not appear to have been addressed previously. Applying an algorithmic approach[20], we find no evidence that hawks attacking swarming bats use closed-loop pursuit of an individual bat––in contrast to raptors attacking singleton targets. Instead, the hawks appear first to turn into the swarm, and then to extend their legs in a grab manoeuvre[18] directed at whichever bat they find themselves on a collision course with as they close range. This could be explained either as an open-loop steering behaviour that involves making an approximately constant radius turn (Fig. 3, S2), or as a closed-loop steering behaviour that involves targeting a spatially fixed point within the swarm (Fig. 5). These alternative hypotheses cannot be distinguished definitively using trajectory data alone, because steering towards a fixed point in space under closed-loop PN guidance at $N = 2$ automatically generates a constant radius turn (Figs. 3 and S2). Nonetheless, the more general hypothesis that the hawks used closed-loop PN guidance at $N \approx 2$ towards a fixed point in the swarm explains more of the data (Fig. 5) and assumes only that whatever guidance mechanism is used to pursue individual targets can also be applied to a virtual target defined as a spatially fixed point within the swarm. Either way, it is important to note that this approach works only in the context of a suitably dense swarm: against a singleton target, the observation that an attacker was turning on a constant radius appropriate to intercept a moving target would be a clear indication that a prediction model was being used. Hence, although we cannot exclude the possibility that a prediction model was used against individuals in the swarm, it is more parsimonious to assume otherwise.

Our finding that the hawks appear either to target a fixed point in space or to follow a predetermined path may in turn explain why neither our present study nor our previous study of the same model system[18] found any evidence to suggest that attacks on the swarm were less successful than attacks on lone bats. This runs counter to the usual assumption that predator hunting efficiency declines as group density increases, owing to the heightened sensory challenge of targeting individual prey[26]. On the contrary, it is this very density that makes it probable that any hawk swooping into the swarm will find itself on a collision course with a bat. This is an appealing interpretation from a sensory perspective, because candidate targets will appear to remain on a constant bearing to an incoming attacker (Fig. 4A, B). The geometric reason for this can be seen by treating the line-of-sight as one side of a curved triangle whose two other sides are formed by the tracks of the predator and prey. If the two individuals are on a direct collision course, then the collision triangle remains geometrically similar through time, such that the line-of-sight remains parallel (Fig. 3). This constant-bearing geometry (Fig. 4) is often taken as evidence of PN guidance[27,28] or motion camouflage[29–31] in dyad interactions, so it is important to emphasize that it holds for any pair of objects that happen to be on a collision course. For instance, the same constant-bearing decreasing-range geometry would arise fortuitously for a bat hit by an arrow shot at random into the swarm. Conversely, objects that are not on course for a collision with each other will usually appear to change their bearing, so drivers learn intuitively to adjust their speed when merging onto a busy freeway, so that the vehicles in the lane they are entering appear on an advancing or receding bearing.

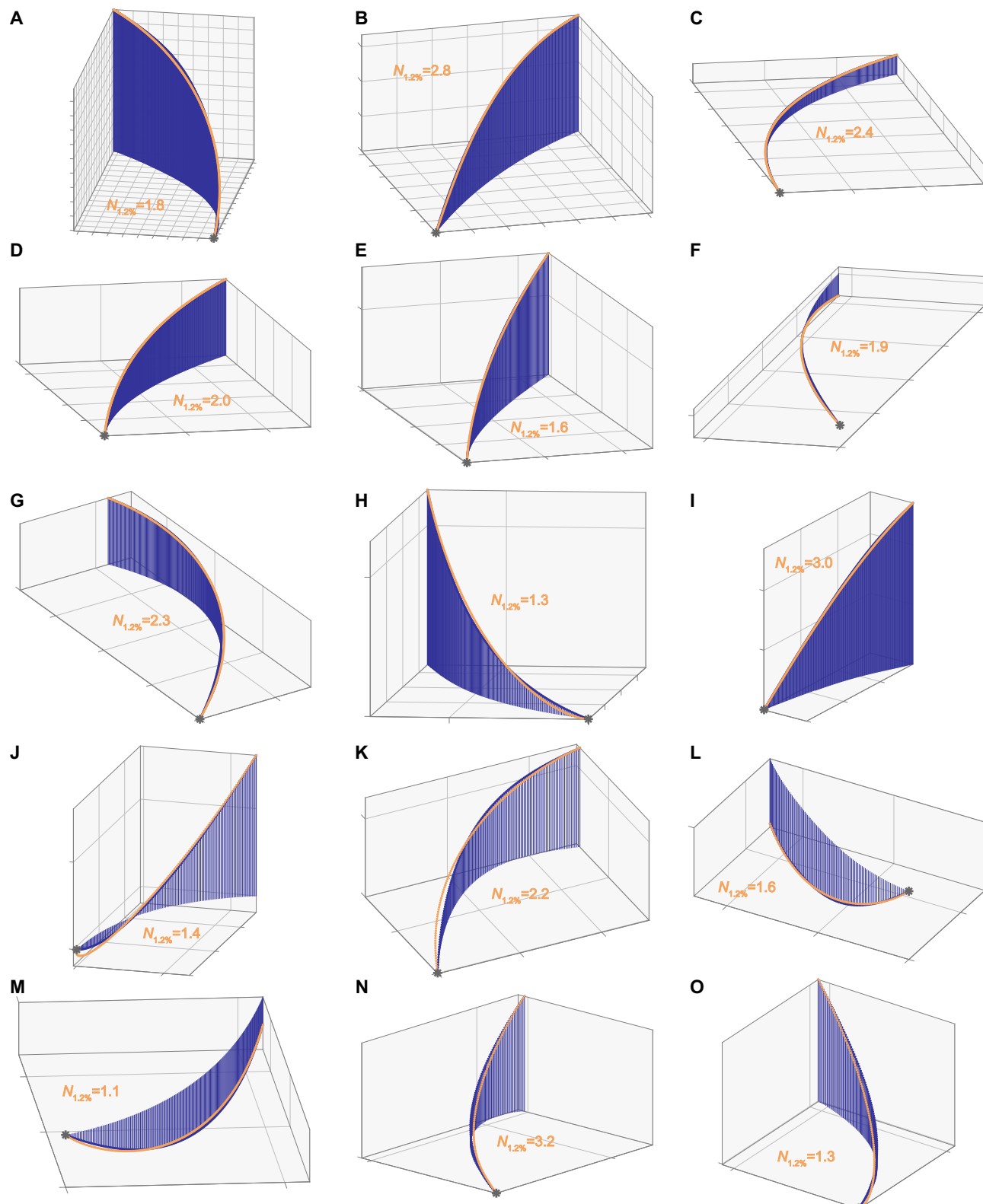

**Fig. 5 | Long-range approaches of Swainson's Hawks attacking Mexican Free-tailed Bats modelled under delay-free PN targeting a fixed point in the swarm.**
**A–O** Each panel plots the reconstructed three-dimensional attack trajectory of an incoming hawk (dark blue points); dark blue lines are dropped vertically from each point to convey the three-dimensional shape of the trajectory; grey starburst shows the point of capture or near-miss. It was not possible to track the bats that the hawks attacked at this range, but orange lines plot simulations of the hawk's flight trajectory generated under delay-free PN guidance at the best-fitting value of the navigation constant $N$, assuming flight at the same speed as measured and treating the final position of the hawk as the target. Trajectories are plotted for the longest section of flight for which the relative error remains below the threshold value of $\varepsilon \leq 0.012$, displaying the subset of 15 flights with the longest simulations meeting this criterion. Grid spacing: 10 m.

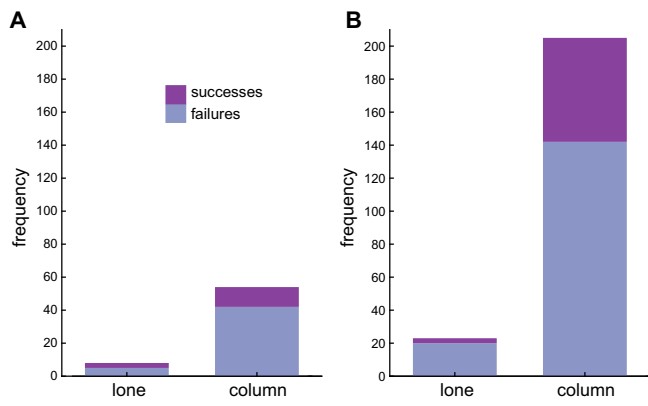

**Fig. 6 | Success rates of hawks attacking lone bats flying outside the column versus swarming bats flying within the column. A** Data from this study. **B** Data from previous study[18] at same field site. In neither case was the hawks' success rate significantly higher when attacking lone bats than when attacking bats flying within the column, but the hawks nevertheless do attack lone bats disproportionately often relative to their frequency in the population[18]. There is therefore no evidence for any confusion effect, but bats flying within the swarm do benefit from attack abatement. See main text for statistical analysis and discussion. Source data are provided as a Source Data file.

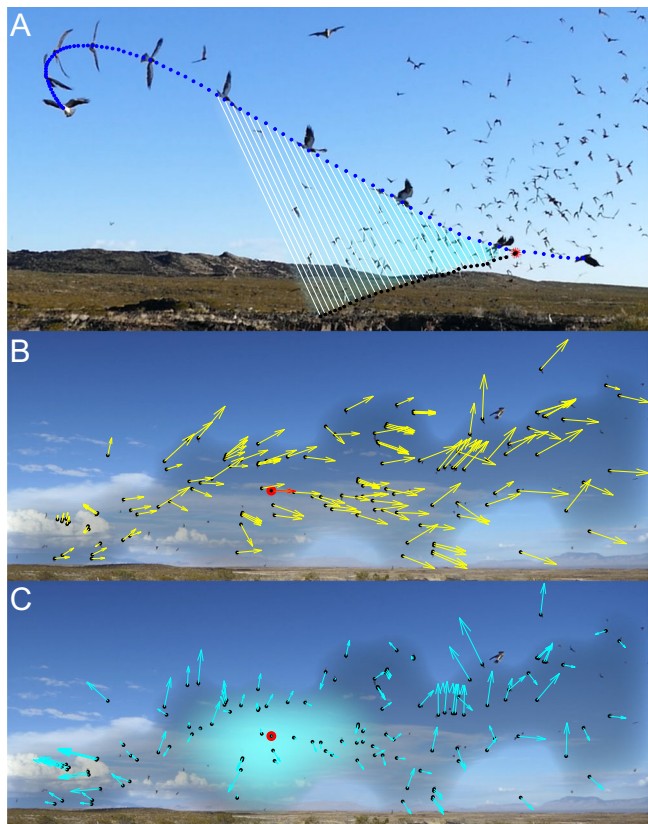

**Fig. 7 | Mechanism by which the constant bearing geometry of an incidental collision course can avoid a confusion effect in a dense swarm. A** Composite image showing the motion of a hawk and the bat that it catches from the swarm, as seen from a fixed video camera. The trajectories of the hawk (blue points) and the bat that it ultimately captures (black points) are sampled at 0.02 s intervals and are shown connected by the instantaneous line-of-sight from the hawk to the bat (white lines). The composite images of the hawk are sampled from the video at 0.2 s intervals and superimposed on the frame corresponding to the catch; red starburst shows the actual point of capture. Note that the line-of-sight remains approximately parallel as the hawk plunges into the swarm, which is because the hawk is on a collision course with the bat that it catches. **B** Projected flight velocity (yellow arrows) of the bats tracked in Fig. 1A (black points), as seen by a stationary observer. The projected flight velocity of the highlighted bat (red arrow) is used as a reference in the next panel. **C** Apparent velocity (cyan arrows) of the same bats, as seen by a moving observer whose translational motion makes the apparent velocity of the highlighted bat (red circle) zero, such that it appears stationary against the distant background. Provided that its range is decreasing simultaneously, this constant bearing geometry means that the observer must be on a collision course with the highlighted bat. Hence, whereas (**B**) represents the optic flow seen by a stationary observer, (**C**) represents the optic flow seen by a moving observer that happens to be on a collision course with the highlighted bat. The constant bearing geometry of an incidental collision course thereby simplifies the sensory challenge of identifying candidate targets in attacks on dense swarms. Darker shading and cyan highlights in (**B**, **C**) indicate the extent of the swarm and highlight the targeted region within it. All frames shown are from the original video collected by the authors in this study.

A similar heuristic could simplify the sensory challenge of selecting a target from within a swarm (Fig. 7). For example, if the hawks were to stabilize their gaze inertially against the distant background, then the retinal position of any candidate target would remain approximately constant, whereas other bats would appear to move against the background. The visual confusion that we experience as stationary observers when viewing a swarm is therefore at least partly resolved for a mobile predator by the constant-bearing geometry of a collision (Fig. 7A). Specifically, whereas all the members of a swarm may appear to move coherently from the viewpoint of a stationary observer (Fig. 7B), any member that is on a collision course with a mobile observer will appear stationary against the residual movement of the swarm (Fig. 7C). It follows that the motion of a swarm will look fundamentally different to a moving observer (Fig. 7C) than to a stationary observer (Fig. 7B), and that this difference aids in target selection and capture. In conclusion, our results show that the confusion effect may be less important to predators attacking dense swarms than has previously been assumed[1], and could also explain why some visually guided predators, including raptors attacking bats[32], may preferentially attack larger or denser groups of prey. This is because plunging into a dense swarm makes it highly probable that an attacker will find itself on a collision course with a prey individual, and the geometry of a collision course means that candidate targets will fall on a constant bearing, making them comparatively straightforward to identify. Hence, collective behaviours that appear confusing from a human standpoint—from swarming bats[18] to murmurating birds[33,34], schooling fish[35–37], and herding ungulates[38]—need not be so confusing to predators taking the plunge.

In conclusion, the reliability of the bats' emergence and the hawks' attacks makes this a useful model system for studying the dynamics of predator-prey interactions in swarms. Reconstructing their flight trajectories using 3D videogrammetry has enabled us to apply algorithmic analysis of pursuit interactions to the natural collective behaviour of wild predators and their wild prey. Our results reveal how dense prey aggregations that appear bewildering to our own eyes will not necessarily result in a confusion effect for mobile predators, which can instead exploit the constant-bearing geometry of a collision course to spot a suitable target as they plunge towards a fixed location within the swarm. This mechanism of collision detection is quite general, so

we expect it to apply to other raptorial predators attacking dense flocks, schools, or swarms in aerial, aquatic, or terrestrial environments. It therefore has significant implications both for our understanding of the adaptive benefits of collective behaviour, and for our understanding of how predators intercept individual prey within a swarm. Since the success of this mechanism hinges upon the presence of a suitably dense prey aggregation, the opportunistic behaviours that we have observed also serve to illustrate how behavioural plasticity is shaped by ecological context and prey behaviour. Finally, our results

have broader implications for the interception of swarming agents such as drones.

## Methods

### Study system

We conducted our research at the Jornada Caves, New Mexico, USA from 8 to 29 June 2018. This remote cave site on private land in the Chihuahuan Desert occupies an elevated volcanic plateau at approximately 1500 m altitude, with the remains of collapsed lava tubes forming a deep canyon with cave and arch features. The site was chosen because of the presence of a population of Swainson's Hawks (*Buteo swainsoni*) that predates the population of Mexican Free-tailed Bats (*Tadarida brasiliensis*) that emerge from the caves en masse daily throughout the summer[39]. The bats migrate to the site during their breeding season from May to September[40], and use the caves as a day roost before flying to their feeding grounds at dusk. The population consists of a maternal colony of approximately 700,000 to 900,000 bats which inhabit two connected caves named North and South. The largest and most reliable emergence was from the South cave, occurring every evening without exception. Emergence from the North cave was less reliable, with no bats emerging at all on some nights during the first week of observations. The numbers of bats were topped up in the second week by new arrivals, and emergence from the North cave was reliable thereafter. Emergence began at a variable time between approximately 18:30 and 20:00 MDT and lasted from 10 to 25 min depending on the number of bats emerging. Sunset was between 20:16 and 20:21 MDT, so the bats usually emerged in broad daylight. During the third week of observations, a substantial second emergence usually occurred at each cave, beginning around 0.5 h after the end of the first emergence, when fewer hawks were present. No ethical issues were identified by the Animal Welfare and Ethical Review Board of the University of Oxford's Department of Zoology. We attended only as observers, and never entered the caves, so the risk of causing disturbance as the bats emerged was low[41].

### Video observations

We recorded video of the hawks attacking the bats every evening from 8 to 29 June 2018, except for one evening that had to be missed due to bad weather. We used three pairs of high-definition video cameras (Lumix DMC-FZ1000/2500, Panasonic Corporation, Osaka, Japan) to enable reconstruction of the three-dimensional flight trajectories of the hawks and bats, setting the camera lens to its widest zoom setting. We recorded 25 Hz video at 3840 × 2160 pixels for the first three days and 50 Hz video at 1920 × 1080 pixels for the remainder of the study (Movie S1). This higher frame rate proved necessary to facilitate tracking of the bats' erratic movements but was traded off against lower spatial resolution. Each camera pair was set in widely spaced stereo configuration to enable three-dimensional reconstruction of the attacks, with a baseline distance of 16 to 27 m. The cameras were mounted on tripods which were adjusted to the same height using an optical level kit (GOL20D/BT160/GR500, Robert Bosch GmbH, Gerlingen, Germany). We used the same optical level kit to measure the baseline distance between the cameras.

We set up two camera pairs facing approximately north and south across the South cave for the duration of the study. As the swarm's overall flight direction was variable and influenced by the wind, we positioned the north- and south-facing camera pairs to allow them to be panned from northeast to northwest and from southeast to southwest, respectively. This enabled us to cover most flight directions, except due east (where the bats rarely flew) and due west (which was subject to glare). We set up a third camera pair to view the emergence that occurred from the North cave from the second week onward. When leaving the North cave, the bats usually flew along the lava tube and beneath a rock arch before climbing out of the canyon.

We therefore positioned the cameras close to where the swarm began climbing out above the canyon rim, aiming to capture attacks as the hawks swooped low over the canyon.

The hawks consistently appeared within a few minutes of the start of emergence, which enabled us to observe the general direction in which the bat swarm was emerging, and to reorient the cameras to view the swarm before the attacks began. As soon as the bats began emerging, the cameras were turned on and left to record. To begin with, all fieldworkers retreated into make-shift hides, but these were gradually phased out for reasons of practicality. The birds quickly became habituated to our presence, venturing close to the cave even when fieldworkers were present. Each attack began with the hawk approaching the swarm in level flight or stooping in from above. This was followed by fast flight through the stream of bats, with one or more attempts made to grab a bat using a pitch-up, pitch-down, or rolling grab manoeuvre with the legs and talons extended (Movie S1). If the first attack was unsuccessful, then the hawks would usually perform further short-range swoops through the stream until they made a catch. Once a bat was caught, the hawk would drift away from the swarm, to consume its prey on the wing.

### Videogrammetry

We synchronized the videos using the DLTdv5 video tracking toolbox[42] in MATLAB R2020a (MathWorks Inc., Natick, MA). To do so, we matched the complex motions involved in the hawks' attack manoeuvres visually between videos, and applied the relevant frame offset to synchronize them to the nearest frame. To verify the accuracy of this method, we compared the position of the hawk's wings between the two videos for the three pairs of frames used for synchronization, and again for the three pairs of frames recorded 50 frames later (Fig. S3). This comparison shows that the frame synchronization remains stable as expected over this 1 s time interval, for the randomly selected flight displayed in Fig. S3. Nevertheless, because the cameras' shutters were not electronically synchronized, this *post hoc* procedure can only guarantee synchronization of the frames to within ±0.01 s at the 50 Hz frame rate (see Fig. S3). To assess the sensitivity of our trajectory reconstructions to this remaining synchronization error, we compared the flight trajectories that we had already reconstructed with those that would have been reconstructed had the videos been shifted ±1 frame (Fig. S4). This comparison shows that the displacement of the trajectories resulting from a synchronization error of ±1 frame is small in comparison to their path length, and that their shape remains approximately the same, even for the two stooping flight trajectories plotted in Fig. S4.

We used the DLTdv5 toolbox to identify the pixel coordinates of the hawk in both videos within a pair, manually tracking the visual centre of the subject's body from the point at which it appeared in both cameras up to the point of interception. We used the same method to track the bat that the hawk caught or attempted to catch during the terminal attack sequences that we recorded at close range. The bats were too distant to be tracked individually in recordings of the hawks' long-range approaches, but the point of actual or attempted capture was nevertheless obvious from the hawks' flight behaviour. We aimed to reconstruct all attack trajectories that were captured by both cameras within a pair. We were able to reconstruct $n = 62$ terminal attack trajectories, drawn from $n = 50$ separate attack flights (i.e. $n = 12$ of these comprised follow-on attack passes, up to a maximum of four consecutive passes made in cases where the first attack pass was unsuccessful; see Supporting Data and Code for details). We were also able to reconstruct $n = 26$ long-range approaches. Hence, as the population of hawks peaked at approximately 20 birds, there will have been repeated sampling within individuals in both cases.

We calibrated the cameras by matching 15 points across both frames, including background features and points on the hawk,

which we selected with the objective of covering as much of the capture volume as possible. The image coordinates of these calibration points were exported from the DLTdv5 toolbox into custom-written software in MATLAB, which solved the camera collinearity equations[43] using a nonlinear least squares bundle adjustment implemented using the MATLAB Optimization Toolbox R2020a (see Supporting Data and Code). The bundle adjustment routine identifies jointly optimal estimates of the camera calibration parameters and unknown spatial coordinates of the calibration points, by minimizing the sum of the squared reprojection error of the associated image points. The reprojection error of an image point matched across camera views is defined as the difference between its measured image coordinates and those expected under the camera calibration model given its estimated spatial coordinates. This nonlinear approach enabled us to self-calibrate the cameras using identified features of the environment, whilst also incorporating prior knowledge of the intrinsic and extrinsic camera parameters. This in turn avoided the need to move a known calibration object through the very large imaging volume.

We set the calibrated baseline distance between the cameras equal to the measurement that we made of this in the field using the optical level. We fixed the focal length of each camera at 1468.9 pixels for the 1920 × 1080 recordings and at 3918.5 pixels for the 3840 × 2160 recordings. These values were estimated using the Camera Calibrator toolbox in MATLAB, from a set of 20 calibration images of a checkerboard pattern held in front of the camera. Lens distortions were found to be minimal, and we therefore assumed a central perspective projection[43] in which we assumed no lens distortion and no principal point offset with respect to the camera sensor. The resulting stereo camera calibration was used to solve for the spatial coordinates of the tracked hawk and bat in MATLAB. This is a least squares solution, in the sense that it minimizes the sum of the squared reprojection error for each image point matched across stereo video frames. We therefore report the root mean square (RMS) reprojection error as a check on the accuracy of the calibrations and reconstructions.

For the terminal attack trajectories filmed at close range, the mean RMS reprojection error of the 16 calibrations was $0.73 \pm 0.35$ pixels, whilst for the reconstructed flight trajectories it was $1.22 \pm 1.18$ pixels for the hawks and $1.87 \pm 2.39$ pixels for the bats over all $n = 62$ flights (mean ± SD). For the long-range approaches filmed at a distance, the RMS reprojection error of the 18 calibrations was $0.53 \pm 0.61$ pixels, whilst for the reconstructed flight trajectories it was $1.08 \pm 1.07$ pixels for the hawks over all $n = 28$ flights (mean ± SD). The sub-pixel reprojection error that we achieved in the calibrations is appropriate to the method. The higher reprojection error of the reconstructions is also to be expected, because whereas the bundle adjustment optimizes the camera calibration parameters jointly with the estimated spatial coordinates of the calibration points, the calibration is held fixed in the reconstructions. In addition, any spatiotemporal error in the matching of points across camera frames will manifest itself as reprojection error in the reconstructions.

The foregoing calibration reconstructs the spatial coordinates of the matched image points in a Cartesian coordinate system aligned with the sensor axes of one of the cameras. To aid visualization and interpretation of the flight trajectories, we therefore transformed the spatial coordinates of the hawks and bats into an Earth axis system in which the $z$ axis was vertical. To do so, we filmed and reconstructed the ballistic trajectory of a small rock thrown high into the air through the volume of stereo overlap. We identified the image coordinates of the peak of its parabolic path, together with the image coordinates of two flanking points located ±20 or 25 frames to either side. We took the line dropped from the peak of the parabola perpendicular to the line connecting the two flanking points to define the direction of

gravitational acceleration. We then used this to identify the rotation needed to transform the spatial coordinates of the hawks and bats into Earth axes with the $z$ axis as vertical. Finally, we made use of the fact that the two cameras in each pair were fixed at the same height to verify the transformation to Earth axes. For the 16 calibrations used to reconstruct the terminal attack trajectories, the inclination of the baseline between the cameras in Earth axes had a median absolute value of just 1.2˚ (1st, 3rd quartiles: 0.8˚, 2.2˚), providing independent validation of the calibration method that we used.

## Trajectory analysis

All trajectory analysis was done using custom-written software in MATLAB R2020a (see Supporting Data and Code). We used piecewise cubic Hermite interpolation of the reconstructed trajectories to estimate the spatial coordinates of the hawk or bat for any occasional frames in which this was obscured. We then smoothed the trajectories using quintic spline fitting. For the long-range approaches, we used a spline tolerance designed to remove an RMS spatial position error of 0.5 m, corresponding approximately to the wing length of a hawk. For the terminal attack trajectories, we used a tolerance designed to remove an RMS position error of 0.12 m, corresponding approximately to the wing length of a bat. These values were chosen as representative estimates of the accuracy with which it was possible to match points across frames at long and close range, respectively. Finally, we differentiated and evaluated the splines analytically to estimate the velocity and acceleration of the bird and bat at an up-sampled frequency of 2 kHz. This ensured a suitably small integration step size for the subsequent numerical simulations. On average, the hawks flew faster than the bats (Fig. S5A), so were tracked over longer distances (Fig. S5B), but with considerable overlap in their respective distributions.

We simulated the hawk's attack trajectory in the Earth axes using a guidance law of the form:

$$\mathbf{a}(t) = N\boldsymbol{\omega}(t - \tau) \times \mathbf{v}(t) - K\boldsymbol{\delta}(t - \tau) \times \mathbf{v}(t) \qquad (1)$$

where $\mathbf{a}$ is the hawk's commanded centripetal acceleration, $\mathbf{v}$ is its velocity, $\boldsymbol{\omega}$ is the angular velocity of the line-of-sight $\mathbf{r}$ from the hawk to its target, and $\boldsymbol{\delta}$ is the deviation angle between $\mathbf{r}$ and $\mathbf{v}$, written in vector form with $\boldsymbol{\delta}$ mutually perpendicular to $\mathbf{r}$ and $\mathbf{v}$. Here, $t$ is time, $\tau$ is a fixed time delay, and $N$ and $K$ are guidance constants. With $K = 0$, Eq. 1 describes proportional navigation (PN), whereas with $N = 0$, Eq. 1 describes pure proportional pursuit (PP). In the case that $K \neq 0$ and $N \neq 0$, Eq. 1 describes mixed PN + PP guidance. Dividing through by the hawk's speed $v = |\mathbf{v}|$ converts the commanded centripetal acceleration to the commanded angular velocity. It can therefore be seen that Eq. 1 generalizes, in vector form, the PN + PP guidance law that is written as $\dot{\gamma}(t) = N\dot{\lambda}(t - \tau) - K\delta(t - \tau)$ in the main text, where the magnitudes of the scalar turn rate, scalar line-of-sight rate, and scalar deviation angle are given respectively as $|\dot{\gamma}| = |\mathbf{a}|/|\mathbf{v}|$, $|\dot{\lambda}| = |\boldsymbol{\omega}|$, and $|\delta| = |\boldsymbol{\delta}|$.

Our simulations make use of the kinematic equations:

$$\mathbf{r} = \hat{\mathbf{x}}_T - \mathbf{x} \qquad (2)$$

$$\boldsymbol{\omega} = \frac{\mathbf{r} \times (\hat{\mathbf{v}}_T - \mathbf{v})}{|\mathbf{r}|^2} \qquad (3)$$

$$\boldsymbol{\delta} = \left( \cos^{-1} \frac{\mathbf{r} \cdot \mathbf{v}}{|\mathbf{r}| \, |\mathbf{v}|} \right) \left( \frac{\mathbf{r} \times \mathbf{v}}{|\mathbf{r} \times \mathbf{v}|} \right) \qquad (4)$$

where $\mathbf{x}$ is the simulated position of the hawk, and where $\hat{\mathbf{x}}_T$ and $\hat{\mathbf{v}}_T$ are the measured position and velocity of the target with respect to the Earth axes. Our simulations are implemented in discrete time

by coupling the guidance law (Eq. 1) with the kinematic equations (Eqs. 2–4) using the difference equations:

$$\mathbf{x}_{n+1} = \mathbf{x}_n + \Delta t\, \mathbf{v}_n. \qquad (5)$$

$$\mathbf{v}_{n+1} = \hat{v}_{n+1}\, \frac{\mathbf{v}_n + \Delta t\, \mathbf{a}_n}{|\mathbf{v}_n + \Delta t\, \mathbf{a}_n|} \qquad (6)$$

where the subscript notation indicates the values of the variables at successive time steps, such that $t_{n+1} = t_n + \Delta t$, and where $\hat{v}$ is the hawk's measured groundspeed. The simulations were initiated given the hawk's measured initial position $\mathbf{x}_0 = \hat{\mathbf{x}}_0$ and velocity $\mathbf{v}_0 = \hat{\mathbf{v}}_0$, and were used to predict the trajectory that it would follow under the guidance law (Eq. 1) parameterized by the guidance constants $N$ and $K$, and time delay $\tau$. Note that Eq. 6 matches the hawk's simulated groundspeed $v = |\mathbf{v}|$ to its measured groundspeed $\hat{v}$ at all times, such that the guidance law is only used to command turning. We verified that the step size of our simulations ($\Delta t = 5 \times 10^{-4}$ s) was small enough to guarantee the numerical accuracy of the fitted guidance parameters and prediction error to the level of precision at which they are reported in the Results.

We defined the prediction error $\eta$ of each simulation as the mean absolute distance between the measured and simulated flight trajectories:

$$\eta = \frac{1}{K} \sum_{n=1}^{k} |\mathbf{x}_n - \hat{\mathbf{x}}_n| \qquad (7)$$

where $\hat{\mathbf{x}}$ is the hawk's simulated position, and $k$ is the number of time steps in the simulation. We fitted the guidance constants $K$ and/or $N$ under the various combinations of guidance law (i.e. PN, PP or PN + PP) and target definition (i.e. measured bat position, final bat position, final hawk position) for delays of $0 \leq \tau \leq 0.1$ s at 0.02 s spacing corresponding to the inter-frame interval. In each case, we used a Nelder–Mead simplex algorithm in MATLAB to find the value of $K$ and/or $N$ that minimised the prediction error $\eta$ for each flight at the given time delay $\tau$. To ensure that we fitted the same section of flight for all time delays $0 \leq \tau \leq 0.1$ s, we began each simulation from 0.1 s after the first point on the trajectory, and ended the simulation at the time of intercept or near-miss. However, as we found the best-fitting delay to be $\tau = 0$, we subsequently re-fitted the simulations with no delay to begin from the first point on the trajectory and report these simulations in the Results. For the terminal attack trajectories, we took the first point on the trajectory to be the earliest point from which it was possible to track the bat that the hawk caught or attempted to catch, and took the time of intercept or near-miss to be the time at which the measured distance between the hawk and bat was minimal. For the long-range approaches, we tested a range of alternative start points from 1.0 s up to a maximum of 20.0 s before the observed grab manoeuvre, in 0.2 s intervals, to accommodate the fact that the hawk could sometimes be tracked for longer than it appeared to be engaged in directed attack behaviour.

## Statistical analysis

All statistics were computed using MATLAB R2020a. As the hawks could not be individually identified, we were unable to control for repeated measures from the same individual, and therefore treated each attack trajectory as an independent sample. Because the distributions of the model parameters and errors are skewed (Fig. 2), we report their median, denoted using tilde notation, together with a bias-corrected and accelerated bootstrap 95% confidence interval (CI) computed using 100,000 resamples[44]. For robustness, we use two-tailed sign tests to compare their distributions between different guidance models and target definitions. We state sample proportions together with a 95% confidence interval (CI) computed using the Clopper–Pearson method. We used a two-tailed Fisher's exact test to compare the odds of success in attacks on lone bats versus attacks on the swarm. Following our previous observational study[18], bats classified as lone bats were judged to be flying >5 body lengths from their nearest neighbours and/or appeared to be flying in a different direction to the coordinated members of the swarm (Table S3).

## Reporting summary

Further information on research design is available in the Nature Research Reporting Summary linked to this article.

## Data availability

Calibration images, digitized image coordinates, and trajectory reconstructions are available as Supporting Data[45] at https://doi.org/10.6084/m9.figshare.19196966. Source Data are provided with this paper for Figs. 2 and 6. Raw video data are stored locally on account of their size (0.2TB) and will be made available upon reasonable request. Source data are provided with this paper.

## Code availability

Guidance simulations and analysis code are available as Supporting Code[45] at https://doi.org/10.6084/m9.figshare.19196966.

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

## Acknowledgements

We thank Turner Enterprises, Inc. and the Armendaris Ranch for access to and housing at the field location. We thank Morgan Kinniry and Paul Domski for support and assistance in the field. This project has received funding from the European Research Council (ERC) under the European Union's Horizon 2020 research and innovation programme (Grant Agreement No. 682501) to G.T., and from an Office of Naval Research Young Investigator Award N000141612478 to L.K.

## Author contributions

Conceptualization: C.B., L.K., G.T. Formal analysis: C.B., G.T. Funding acquisition: L.K., G.T. Investigation: C.B., L.K., C.H., L.L., K.M., L.Z., G.T. Methodology: C.B., L.K., G.T. Software: C.B., G.T. Supervision: L.K., G.T. Visualization: C.B., G.T. Writing – original draft: C.B., G.T. Writing – review and editing: C.B., L.K., G.T.

## Competing interests

The authors declare no competing interests.
