## [Peer Review File · Nature Communications]

Raptors avoid the confusion effect by targeting fixed points in dense aerial prey aggregationsReviewers' Comments:

Reviewer #1:

Remarks to the Author:

This is an important paper, bringing together research on intercepting targets with the confusion effect, to my knowledge, for the first time. A significant body of research has built up on target interception and the confusion effect independently of one another, thus the study should have strong appeal to both researchers interested in the adaptive value of group living as well as those interested in how targets are intercepted. The study system is ideal for addressing the questions, with the predictability of the emerging bats and the attacks of their avian predators making this kind of study viable. The data collection, analysis and interpretation is robust as far as I can tell. The paper is exceedingly well written, structured and argued, and makes numerous novel points that deserve to influence the direction of research on predators attacking aggregations of prey (the final paragraph, starting on line 388, was particularly interesting). My points for improvement below are very minor.

L34: Regarding marginal predation, the broad Krause and Ruxton book (ref 7) should be replaced by a primary literature study on the subject, e.g.

The measure of spatial position within groups that best predicts predation risk depends on group movement (2021) PJ Lambert, JE Herbert-Read, CC Ioannou, *Proceedings of the Royal Society B* 288, 20211286

L62-64: Defining / making clear what open and closed loop mean in this context would be useful, given that the journal has a broad readership and those attracted to the paper from an interest in the confusion effect may not be familiar with these terms. The terms are used repeatedly through the main text so it is particularly important they are clear to a wide range of readers.

L220-221: Given the small size of the points, I found differentiating between the dark grey and dark blue difficult.

L344: As with the comment above, the text here ("an Eulerian rather than Lagrangian approach to targeting") is likely to be unclear to many readers.

L372: As above, there are papers much more specifically on this point than the general Krause and Ruxton book, e.g.

The confusion effect—from neural networks to reduced predation risk (2008) CC Ioannou, CR Tosh, L Neville, J Krause, *Behavioral Ecology* 19 (1), 126-130

L406: For a very recent and elegant schooling fish example, see:

Fish waves as emergent collective antipredator behavior (2022) C Doran et al. *Current Biology* 32 (3), 708-714. e4

Christos Ioannou
University of Bristol

Reviewer #2:

Remarks to the Author:

This paper reports on observed pursuit behaviour of raptors preying against swarms of bats. Hypotheses about guidance laws that might be in use were applied, finding that the most likely hypothesis was an absence of guidance law, other than steering with respect to the centre of the swarm.

The paper is well written, although might I say a little informal in usage, for example "grabbed",

"fluke" and so on: The former being too informal and non-specific, and the second being slightly uncommon informal usage, better replaced with a more specific term. Line 161, is the correct word "overlain"? or "overlaid"?.

I am happy with the interpretation of the data. I do have a question about the videogrammetry. You report on the reprojection error, and make mention of higher temporal resolution being preferred to improve trajectory measurements. This is related to the ability to reconstruct a static point in the scene. What is the dynamic effect due to lack of frame synchronisation? what error does it cause? Will it be significant? Some quite high derivatives are being taken from these measurements. I suspect that everything is simply offset, but I could be wrong. Also worth commenting on how stable the time-base is, how far out of sync were the cameras and how close to the same frequency, mainly since the topic comes up in line 480. If they were a long way out of sync, the time offset might be evolving during the flight.

Reviewer #3:

Remarks to the Author:

The authors explore steering strategies of predatory hawks targeting swarms of bats. The observations and data analysis are sound and support the authors' conclusions. There's a wealth of research on how predators track targets but this manuscript is novel because in this work the authors aim to evaluate the confusion effect of swarming behavior of prey and its effect on capture success.

I have only minor comments:

1) One of the main findings of the paper, or at least the one driving the title, doesn't have a dedicated figure, all the results are in the text starting line 304. A figure showcasing these findings would be appropriate.

2) The authors do not discuss the possibility that the animals are integrating information to form prediction models on their steering strategy. This has been reported in several predatory species that track erratic prey. In this case, by targeting a fixed point in the swarm, it seems like the hawks are playing a game of chance, and this might be more energetically efficient than predicting motion; but the discussion could do with a mention of this.

Prediction models could explain the capture success that involve trajectories with steeper curves, such as those in figure 3 C, I and N.

3) The discussion could also do with a more generalized view of the significance of this research, especially considering the target audience of the journal.

REVIEWER COMMENTS

Reviewer #1 (Remarks to the Author):

This is an important paper, bringing together research on intercepting targets with the confusion effect, to my knowledge, for the first time. A significant body of research has built up on target interception and the confusion effect independently of one another, thus the study should have strong appeal to both researchers interested in the adaptive value of group living as well as those interested in how targets are intercepted. The study system is ideal for addressing the questions, with the predictability of the emerging bats and the attacks of their avian predators making this kind of study viable. The data collection, analysis and interpretation is robust as far as I can tell. The paper is exceedingly well written, structured and argued, and makes numerous novel points that deserve to influence the direction of research on predators attacking aggregations of prey (the final paragraph, starting on line 388, was particularly interesting). My points for improvement below are very minor.

We thank the reviewer for their positive assessment of our manuscript, and for their time taken in reviewing it. We have implemented all their suggestions as described below.

L34: Regarding marginal predation, the broad Krause and Ruxton book (ref 7) should be replaced by a primary literature study on the subject, e.g. The measure of spatial position within groups that best predicts predation risk depends on group movement (2021) PJ Lambert, JE Herbert-Read, CC Ioannou, Proceedings of the Royal Society B 288, 20211286

We have replaced the reference as suggested.

L62-64: Defining / making clear what open and closed loop mean in this context would be useful, given that the journal has a broad readership and those attracted to the paper from an interest in the confusion effect may not be familiar with these terms. The terms are used repeatedly through the main text so it is particularly important they are clear to a wide range of readers.

We have expanded this section to read: "Attacks on solitary prey necessarily require accurate guidance. Open-loop interception may be possible against a predictably moving target, and means simply that the attacker aims at its target's expected future position, without making further corrections to account for unexpected changes in the target's relative motion (e.g. an archer shooting a moving target with a bow only has control over the arrow's initial release velocity). On the other hand, closed-loop steering is essential when pursuing an evasive target, and means that the attacker uses information on the relative motion of its target to make continuous adjustments to its own flight trajectory (e.g. a heat-seeking missile intercepts its target by tracking its thermal signature)."

L220-221: Given the small size of the points, I found differentiating between the dark grey and dark blue difficult.

The colour of the bat trajectory has been changed from grey to magenta.

L344: As with the comment above, the text here (“an Eulerian rather than Lagrangian approach to targeting”) is likely to be unclear to many readers.

This has been deleted from the text.

L372: As above, there are papers much more specifically on this point than the general Krause and Ruxton book, e.g. The confusion effect—from neural networks to reduced predation risk (2008) CC Ioannou, CR Tosh, L Neville, J Krause, Behavioral Ecology 19 (1), 126-130.

We have replaced the reference as suggested.

L406: For a very recent and elegant schooling fish example, see: Fish waves as emergent collective antipredator behavior (2022) C Doran et al. Current Biology 32 (3), 708-714. e4

We have added the reference as suggested.

Christos Ioannou
University of Bristol

We thank you, Christos, for your time and in reviewing our work, and for your helpful comments, which we have been pleased to implement.

Reviewer #2 (Remarks to the Author):

This paper reports on observed pursuit behaviour of raptors preying against swarms of bats. Hypotheses about guidance laws that might be in use were applied, finding that the most likely hypothesis was an absence of guidance law, other than steering with respect to the centre of the swarm.

We thank the reviewer for their time taken in reviewing our manuscript, and for their positive comments and helpful suggestions – all of which we have implemented.

The paper is well written, although might I say a little informal in usage, for example "grabbed", "fluke" and so on: The former being too informal and non-specific, and the second being slightly uncommon informal usage, better replaced with a more specific term. Line 161, is the correct word "overlain"? or "overlayed"?

We have changed most uses of the words “grab” or “grabbed” to “attempted catch”, “attempted to catch”, “caught”, or “attacked”, according to the context. We have reserved use of the phrase “grab manoeuvre” to refer to the specific manoeuvre involving rapid extension of the legs that we defined as such in our previous work [ref. 18] and which we describe here in the Results and Methods. We have changed “fluking” to “achieving a hit by chance”, and have corrected “overlain” to “overlaid”.

I am happy with the interpretation of the data. I do have a question about the videogrammetry. You report on the reprojection error, and make mention of higher temporal resolution being preferred to improve trajectory measurements. This is related to the ability to reconstruct a static point in the scene. What is the dynamic effect due to lack of frame synchronisation? what error does it cause? Will it be significant? Some quite high derivatives are being taken from these measurements. I suspect that everything is simply offset, but I could be wrong. Also worth commenting on how stable the time-base is, how far out of sync were the cameras and how close to the same frequency, mainly since the topic comes up in line 480. If they were a long way out of sync, the time offset might be evolving during the flight.

This is an important question. The dynamic effect of any errors in frame synchronization is small, as shown by the two new supplementary figures that we have created to demonstrate this (new Figs. S3-4). We have added the following explanation to the Methods:

“To verify the accuracy of this method, we compared the position of the hawk’s wings between the two videos for the three pairs of frames used for synchronization, and again for the three pairs of frames recorded 50 frames later (Fig. S3). This comparison shows that the frame synchronization remains stable as expected over this 1 s time interval, for the randomly selected flight displayed in Fig. S3. Nevertheless, because the cameras’ shutters were not electronically synchronized, this *post hoc* procedure can only guarantee synchronization of the frames to within ± 0.01 s at the 50 Hz frame rate (see Fig. S3). To assess the sensitivity of our trajectory reconstructions to this remaining synchronization error, we compared the flight trajectories that we had already reconstructed with those that would have been reconstructed had the videos been shifted ± 1 frame (Fig. S4). This comparison shows that the displacement of the trajectories resulting from a synchronization error of ± 1 frame is small in comparison to their path length, and that their shape remains approximately the same, even for the two stooping flight trajectories plotted in Fig. S4.”

Reviewer #3 (Remarks to the Author):

The authors explore steering strategies of predatory hawks targeting swarms of bats. The observations and data analysis are sound and support the authors’ conclusions. There’s a wealth of research on how predators track targets but this manuscript is novel because in this work the authors aim to evaluate the confusion effect of swarming behavior of prey and its effect on capture success.

We thank the reviewer for their time taken in reviewing our manuscript, and for their positive comments and helpful suggestions – all of which we have implemented.

I have only minor comments:

1) One of the main findings of the paper, or at least the one driving the title, doesn't have a dedicated figure, all the results are in the text starting line 304. A figure showcasing these findings would be appropriate.

Thank you for this helpful suggestion. We have added a new figure (new Fig. 6) displaying this information as recommended.

2) The authors do not discuss the possibility that the animals are integrating information to form prediction models on their steering strategy. This has been reported in several predatory species that track erratic prey. In this case, by targeting a fixed point in the swarm, it seems like the hawks are playing a game of chance, and this might be more energetically efficient than predicting motion; but the discussion could do with a mention of this. Prediction models could explain the capture success that involve trajectories with steeper curves, such as those in figure 3 C, I and N.

Thank you for this helpful suggestion. We have added the following sentences to the Results:

“whilst it is possible in principle that an attacker could calculate the radius of turn needed to reach a fixed point in space, doing so requires an estimate of target range, which may be lacking given the absence of any evidence for stereopsis in diurnal raptors²⁴. The same will hold true of any prediction model in which the future position of an individual target is predicted explicitly. We are therefore left with two alternative hypotheses that are equally consistent with the data: first, that the hawks enter a turn of approximately constant radius, with or without knowledge of where they will intercept the swarm; second, that the hawks use PN guidance at $N \approx 2$ to target a fixed point in space, without knowledge of what radius of turn they will follow. These hypotheses each represent very different mechanisms: whereas the first describes a form of open-loop steering behaviour, with or without prediction, the second describes a form of closed-loop guidance from which a similar-looking behaviour emerges.”

and have picked the point up again in the Discussion, where we write:

“the hawks appear first to turn into the swarm, and then to extend their legs in a grab manoeuvre¹⁸ directed at whichever bat they find themselves on a collision course with as they close range. This could be explained either as an open-loop steering behaviour that involves making an approximately constant radius turn (Figs. 3, S2), or as a closed-loop steering behaviour that involves targeting a spatially fixed point within the swarm (Fig. 5). We cannot distinguish reliably between these hypotheses because steering under closed-loop PN guidance at $N = 2$ automatically generates a constant radius turn (Fig. 3, S2). Nonetheless, the general hypothesis that the hawks used closed-loop steering towards a fixed point in the swarm explains more of the data (Fig. 5), and assumes only that whatever guidance mechanism is used to pursue individual targets can also be applied to a virtual target defined as a spatially fixed point within the swarm. It is important to note that this approach works *only* in the context of a suitably dense swarm: against a singleton target, the observation that an attacker was turning on a constant radius appropriate to intercept a moving target would be a clear indication that a prediction model was being used. Hence, although we cannot exclude the possibility that a prediction model was used against individuals in the swarm, it is more parsimonious to assume otherwise.”

3) The discussion could also do with a more generalized view of the significance of this research, especially considering the target audience of the journal.

Thank you for this helpful suggestion. We have added the following closing paragraph as recommended:

“In conclusion, the reliability of the bats’ emergence and the hawks’ depredation behaviour makes this an excellent study system for addressing questions on predator-prey interactions in swarms. This study is, to our knowledge, the first to reconstruct and analyse the 3D flight trajectories of wild predators and their wild prey, and the first to apply algorithmic analysis of predator pursuit interactions to collective behaviour. Our results reveal how dense prey aggregations that appear bewildering to our own eyes will not necessarily result in a confusion effect for mobile predators, which can instead exploit the constant-bearing geometry of a collision course to spot a suitable target as they plunge towards a fixed location within the swarm. This mechanism of collision detection is completely general, so we may expect it to apply to other raptorial predators attacking dense flocks, schools, or swarms in aerial, aquatic, or terrestrial environments. It therefore has significant implications both for our understanding of the adaptive benefits of collective behaviour, and for our understanding of how predators intercept individual prey within a swarm. Since the success of this mechanism hinges upon the presence of a suitably dense prey aggregation, the opportunistic behaviours that we have observed also serves to illustrate how behavioural plasticity is shaped by ecological context and prey behaviour. Finally, our results have broader implications for the interception of swarming agents such as drones.

Reviewers' Comments:

Reviewer #2:

Remarks to the Author:

I am satisfied with the authors' responses and with the changes to the manuscript that they have proposed.

Reviewer #3:

Remarks to the Author:

The authors have addressed all my comments and I congratulate them on an outstanding manuscript.